Review

 

Subject Area:
genetics/biochemistry

Keywords:
vitamin B₆, genome integrity, cancer, *Drosophila melanogaster*

Authors for correspondence:
Angela Tramonti
e-mail: angela.tramonti@cnr.it
Fiammetta Vernì
e-mail: fiammetta.verni@uniroma1.it

# The multifaceted role of vitamin B₆ in cancer: *Drosophila* as a model system to investigate DNA damage

Roberto Contestabile[1], Martino Luigi di Salvo[1], Victoria Bunik[2,3,4], Angela Tramonti[1,5] and Fiammetta Vernì[6]

[1]Istituto Pasteur Italia-Fondazione Cenci Bolognetti and Dipartimento di Scienze Biochimiche 'A. Rossi Fanelli', Sapienza Università di Roma, P.le A. Moro, 5, 00185, Roma, Italy
[2]Belozersky Institute of Physico-Chemical Biology, and [3]Faculty of Bioengineering and Bioinformatics, Lomonosov Moscow State University, Moscow 119991, Russia
[4]Sechenov Medical University, Sechenov University, 119048 Moscow, Russia
[5]Istituto di Biologia e Patologia Molecolari, Consiglio Nazionale delle Ricerche, Pl.e A. Moro, 5, 00185 Roma, Italy
[6]Dipartimento di Biologia e Biotecnologie 'Charles Darwin', Sapienza Università di Roma, Pl.e A. Moro, 5, 00185, Roma, Italy

RC, 0000-0002-5235-9993; MLdS, 0000-0003-1233-9338; AT, 0000-0002-5625-1170; FV, 0000-0001-8866-3324

A perturbed uptake of micronutrients, such as minerals and vitamins, impacts on different human diseases, including cancer and neurological disorders. Several data converge towards a crucial role played by many micronutrients in genome integrity maintenance and in the establishment of a correct DNA methylation pattern. Failure in the proper accomplishment of these processes accelerates senescence and increases the risk of developing cancer, by promoting the formation of chromosome aberrations and deregulating the expression of oncogenes. Here, the main recent evidence regarding the impact of some B vitamins on DNA damage and cancer is summarized, providing an integrated and updated analysis, mainly centred on vitamin B₆. In many cases, it is difficult to finely predict the optimal vitamin rate that is able to protect against DNA damage, as this can be influenced by a given individual's genotype. For this purpose, a precious resort is represented by model organisms which allow limitations imposed by more complex systems to be overcome. In this review, we show that *Drosophila* can be a useful model to deeply understand mechanisms underlying the relationship between vitamin B₆ and genome integrity.

## 1. Impact of most representative B group vitamins on DNA damage and cancer: *in vitro* and *in vivo* studies

The study of micronutrients is a topic of general interest, due to the impact of minerals and vitamins on human health. Growing evidence shows that the deficiency of several vitamins causes DNA damage predisposing to cancer and neurological diseases, but cause–effect relationships in most of the cases are not completely understood. Many micronutrients work as cofactors or substrates for enzymes that counteract genotoxins or are involved in DNA metabolism, and their deficiency can damage DNA analogously to common carcinogens [1]. In many cases, it is difficult to finely predict the optimal rate of micronutrients that is able to protect against DNA damage, as this rate can be influenced by the individual's genotype [2]. Thus, the need arises to explore in depth the pleiotropic action and the metabolism of vitamins, in order to set supportive interventions and personalized cares.

Vitamins B₉, B₁₂, B₁ and B₆ (dietary sources reported in table 1) are the source of coenzymes that participate in one carbon metabolism, in which 1C units are

royalsocietypublishing.org/journal/rsob Open Biol. 10: 200034

**Table 1.** Dietary sources and recommended daily allowance for vitamins $B_1$, $B_6$, $B_9$ and $B_{12}$ (from https://www.ncbi.nlm.nih.gov/books/NBK554545/).

| vitamin | dietary sources | RDA (recommended dietary allowance) |
|---|---|---|
| vitamin $B_1$ (thiamine) | found in all foods in moderate amounts: pork, legumes, enriched and whole grains, cereals, nuts and seeds | 1.1 mg day$^{-1}$ for adult women and 1.2 mg day$^{-1}$ for adult men |
| vitamin $B_6$ (pyridoxine) | widespread among food groups: meat, fish, poultry, vegetables, fruits | 1.3 mg day$^{-1}$ for adults |
| vitamin $B_9$ (folic acid) | leafy green vegetables and legumes, liver, seeds, orange juice, enriched and fortified grains | 400 mcg day$^{-1}$ of *dietary folate equivalents*[a] for adults; recommendation is that women of childbearing age consume an additional 400 mcg day$^{-1}$ of folic acid from supplements or fortified foods to decrease the risk of neural tube defects |
| vitamin $B_{12}$ (cobalamin) | only present in animal products because it is a product of bacteria synthesis: meat, poultry, fish, seafood, eggs, milk and milk products; not found in plant foods; many foods are also fortified with synthetic vitamin $B_{12}$ | 2.4 mcg day$^{-1}$ for adults; it is recommended for older adults to meet their RDA with fortified foods or supplements because many are unable to absorb naturally occurring vitamin $B_{12}$ |

[a]Dietary folate equivalents (DFE) take into account the lower availability of mixed folates in food compared with synthetic tetrahydrofolate used in food enrichment and supplements. Currently, the use of DFE is recommended for planning and evaluating the adequacy of people's folate intake.

used in biosynthetic processes such as purine and thymidylate synthesis and homocysteine remethylation (figure 1). Consistently, a large body of evidence shows that deficiency of these vitamins impacts on genome stability and cancer. Vitamin $B_9$ encompasses a group of compounds collectively named as folates, including folic acid, tetrahydrofolic acid (THF; or $H_4$-pteroyl-L-glutamate), 5-methyltetrahydrofolic acid ($CH_3$-THF) and 5,10-methylenetetrahydrofolic acid ($CH_2$-THF), required for growth and development. Dietary folic acid is first reduced to dihydrofolate and then to tetrahydrofolate by the activity of dihydrofolate reductase. Folate deficiency (FD) causes genome instability as assessed by *in vitro* studies on human and animal cell cultures. In particular, FD produces fragile sites [3], chromosome breakage [4] and aneuploidy [5]. Cytokinesis-block micronucleus assays in primary human lymphocyte cultures deprived of folate revealed micronuclei, which contain chromosomes or chromosome fragments not incorporated into one of the daughter nuclei during cell division, nucleoplasmic bridges (a biomarker of dicentric chromosomes resulting from telomere end-fusions or DNA misrepair) and nuclear buds (a marker of gene amplification and/or altered gene dosage) [6].

*In vitro* observations have been complemented with epidemiological [7,8] and controlled intervention studies [9–11], further reinforcing the association between folate and genome stability. Consistently, a growing body of evidence indicates that FD may increase risk for several cancer, including those of colon, pancreas, prostate and breast [12,13]. To explain the effects of FD on genome stability, two mechanisms have been proposed: the impaired conversion of dUMP in dTMP and the hypomethylation of DNA. Folate is required for conversion of deoxyuridine monophosphate (dUMP) to deoxythymidine monophosphate (dTMP) performed by thymidylate synthase (TS) (figure 1). Therefore, FD can cause dUTP incorporation in DNA, instead of dTTP, which is removed by uracil glycosidase, resulting in mutations, chromosome aberrations and eventually cancer. In addition, the unbalanced dUTP/dTTP ratio can impair DNA synthesis and repair, increasing genetic instability. As a confirmation of

this model, treatment of human lymphoid cells in culture with methotrexate, an inhibitor of dihydrofolate reductase, increases the dUTP/dTTP ratio and the rate of uracil misincorporation in DNA [14]. Moreover, *in vitro* folic acid depletion causes uracil misincorporation in human lymphocytes [15].

Folate is also required for the production of *S*-adenosylmethionine (SAM) throughout the remethylation of homocysteine to methionine (figure 1). In turn, SAM regulates gene transcription by methylating specific cytosines in DNA. As a consequence, low folate levels may lead to DNA hypomethylation, which can potentially induce proto-oncogenes expression. An altered methylation pattern has been proposed to be at the basis of aneuploidy caused by FD. According to this model, it has been proposed that demethylation of heterochromatic centromeric regions could impair the correct distribution of chromosomes during nuclear division [16]. However, more recently, it has been proposed that aneuploidy in FD cells can also result from spindle assembly checkpoint (SAC) dysfunction, due to an altered expression of some SAC genes induced by FD [17].

Vitamin $B_{12}$ is essential for maintaining nervous system functions as well as haematopoiesis [18,19]. Suboptimal $B_{12}$ status (serum $B_{12} < 300$ pmol l$^{-1}$) is very common, occurring in 30–60% of the population, in particular in pregnant women and in less-developed countries. Vitamin $B_{12}$, together with folate, serves as coenzyme for methionine synthase (MS) (figure 1). When $B_{12}$ is insufficient, THF is trapped as $CH_3$-THF. This hinders the regeneration of THF and reduces the size of the $CH_2$-THF pool, leading to increased dUTP misincorporation into DNA. Reduced MTR activity increases homocysteine in tissue and plasma, a biomarker associated with several diseases, including risk of neural tube defects [20].

The first evidence that vitamin $B_{12}$ deficiency is associated with chromosome damage in human cells has been the presence of 'Howell–Jolly bodies' in erythrocytes from patients affected by megaloblastic anaemia (a disease caused by $B_{12}$ deficiency). Howell–Jolly bodies are small round nuclear remnants caused by chromosome breakage and chromosome segregation defects, as they contain whole chromosomes or

royalsocietypublishing.org/journal/rsob    *Open Biol.* **10**: 200034

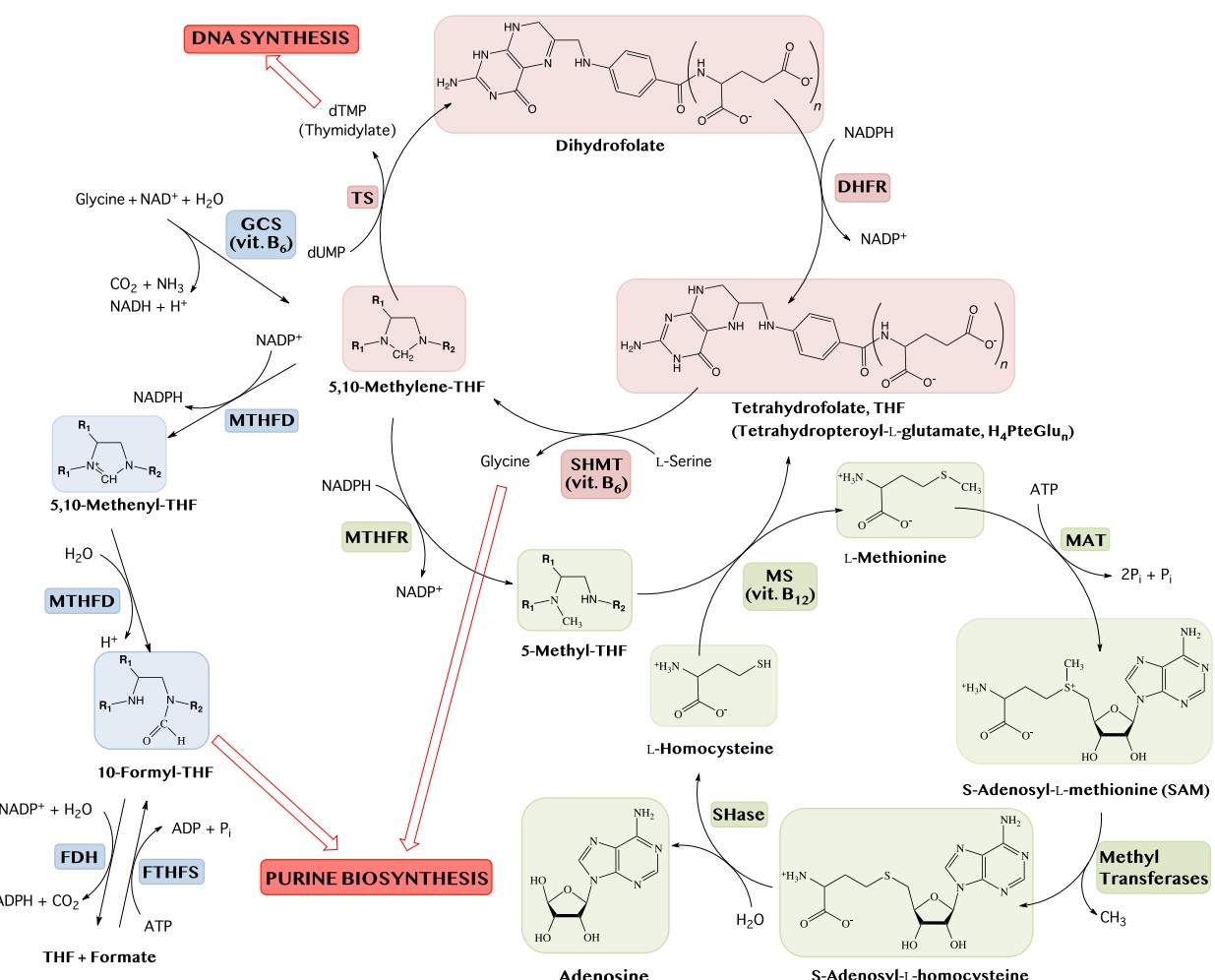

**Figure 1.** Schematic of $B_9$ metabolism comprising the thymidylate cycle (red diagram), the methionine cycle (green diagram) and the purine biosynthesis pathway (blue diagram). The enzymes involved are: dihydrofolate reductase (DHFR); thymidylate synthase (TS); serine hydroxymethyltransferase (SHMT); methylenetetrahydrofolate reductase (MTHFR); methionine synthase (MS); methionine adenosyltransferases (MAT); *S*-adenosylhomocysteinase (SHase); glycine cleavage system (GCS); methylenetetrahydrofolate dehydrogenase (MTHFD); 10-formyltetrahydrofolate dehydrogenase (FDH); formyltetrahydrofolate synthetase (FTHFS).

chromosome fragments that lag behind at anaphase [21]. In line with these findings, subsequent *in vivo* and *in vitro* studies have associated low $B_{12}$ levels with increased chromosome damage, and a significant negative correlation has been demonstrated between micronucleus index and serum vitamin $B_{12}$ content [9,22–24]. Intervention studies showed that DNA damage and micronucleus frequency is significantly improved through vitamin $B_{12}$ therapy [23,25,26].

Although low $B_{12}$ levels are also expected to be associated with cancer, there is only little evidence on this. Indirect evidence come from a study indicating that smokers with low $B_{12}$ levels had high rate of micronuclei, suggesting that low $B_{12}$ levels could be correlated to epithelial cancers [27]. Another study suggested that elevated total $B_{12}$ could be considered as a potential marker for oncohaematological disorders [28].

The coenzyme active form derivative of vitamin $B_1$ (thiamine) is thiamine pyrophosphate (TPP), an essential cofactor of several key enzymes in cellular metabolism, among which is transketolase (TKT) within the pentose phosphate pathway (PPP). Three other phosphorylated forms have been observed intracellularly in humans in addition to TPP: thiamine monophosphate, thiamine triphosphate and adenosine thiamine triphosphate [29]. In cancer cells, TKT within the PPP is responsible for the synthesis of most ribose 5-phosphate (R5P). In normal cells, R5P is produced through the non-

thiamine-dependent oxidative branch of PPP. If an excess of R5P is present with respect to cell requirements, it is recycled into glucose 6-phosphate through the non-oxidative branch of the PPP, in which TKT is present, where R5P is converted to fructose 6-phosphate and glyceraldehyde 3-phosphate. In cancer cells, the large requirement of R5P needed for nucleotide synthesis determines an inversion of the normal metabolic flux, increasing reliance on the non-oxidative branch PPP for R5P production [30]. Accordingly, inhibition of thiamine metabolism is expected to result in the reduction in the nucleotide pools. The thiamine analogue and anti-coenzyme oxythiamine was shown to reduce DNA and RNA synthesis, through reduction in R5P, and therefore tumour cell growth both *in vivo* and *in vitro* [31]. High importance of thiamine in malignant cells was shown in both epidemiological [32] and biochemical [33] studies. In humans, cancer rates correlate with thiamine status [32]. Thiamine depletion of normal tissues due to strong thiamine mobilization by cancer cells [33] may often cause complications in cancer patients, such as heart failure [34].

Thiamine and TPP have also been demonstrated to have antioxidant properties, reacting with ROS [35]. In particular, TPP has provided a greater protective effect against oxidative stress-induced damage (i.e. DNA hydroxylation) compared with thiamine [36].

**Figure 2.** Schematic of vitamin $B_6$ metabolism in humans. The orange diagram corresponds to the pyridoxal 5′-phosphate salvage pathway. PLP, pyridoxal 5′-phosphate; PNP, pyridoxine 5′-phosphate; PMP, pyridoxamine 5′-phosphate; PL, pyridoxal; PN, pyridoxine; PM, pyridoxamine; PA, 4-pyridoxic acid; PDXK: pyridoxal kinase; TNSALP: tissue-non-specific alkaline phosphatases; PLPP, pyridoxal 5′-phosphate phosphatase; ALDH, aldehyde dehydrogenases; POX, pyridoxal oxidase; AOX, aldehyde oxidases.

**Table 2.** Biological functions of $B_6$ vitamers.

| $B_6$ vitamer | function | reference |
|---|---|---|
| PLP and PMP | catalysis (enzyme cofactor) | [37] |
| PLP and PN | binding to steroid receptors, playing a role in membrane transport | [38–40] |
| all vitamers | reactive oxygen species scavenger and resistance factor to biotic and abiotic stress in plants and in *Plasmodium falciparum* | [41–44] |
| PLP | virulence factor in *Helicobacter pylori*, *Mycobacterium tuberculosis* and *Actinobacillus pleuropneumoniae* | [45–47] |
| PLP | chaperone in enzyme folding | [48] |
| PLP | modulator of transcription factors | [49,50] |
| PLP and PMP | inhibition of the formation of advanced glycation end products (AGEs) | [51–53] |

# 2. Roles of vitamin $B_6$ in human health and disease

## 2.1. Vitamin $B_6$ metabolism in humans

Vitamin $B_6$ is an ensemble of six substituted pyridine compounds or vitamers: pyridoxine (PN), pyridoxal (PL), pyridoxamine (PM) and their related 5′-phosphate derivatives (figure 2). The catalytically active form of the vitamin, pyridoxal 5′-phosphate (PLP), acts as a cofactor for over 150 enzymes [37] involved in a number of crucial metabolic pathways, such as the synthesis, transformation and degradation of amines and amino acids,

supply of one carbon units, transsulfuration, synthesis of tetrapyrrolic compounds (including haem) and polyamines, biosynthesis and degradation of neurotransmitters. Moreover, $B_6$ vitamers are involved in important biological functions other than catalysis (table 2).

The different $B_6$ vitamers are interconverted through a salvage pathway that involves pyridoxal kinase (PDXK), pyridoxine 5′-phosphate oxidase (PNPO) and several phosphatases (figure 2). The ATP-dependent PDXK phosphorylates the 5′ alcohol group of PN, PL and PM to form PNP, PLP and PMP, whereas the FMN-dependent PNPO oxidizes PNP and PMP to give PLP. Tissue-non-specific alkaline phosphatase (TNSALP) is a lipid-anchored ectophosphatase present on the external

royalsocietypublishing.org/journal/rsob    Open Biol. **10**: 200034

surface of the cell membrane in several organs such as liver, bone and kidney. Its physiological role is to dephosphorylate $B_6$ vitamers so as to allow their transport across the membrane [54]. On the other hand, an intracellular, cytosolic PLP phosphatase exists, which is ubiquitously expressed in humans and is specifically involved in vitamin $B_6$ catabolism [55].

Another important component of vitamin $B_6$ metabolism is the recently discovered PLP-binding protein (PLP-BP), widespread in all kingdoms of life, with no catalytic activity but with an important regulatory function in PLP homeostasis [56]. In fact, PLP is a very reactive aldehyde that easily combines with amino and thiol groups. Therefore, it is important to maintain a correct balance among $B_6$ vitamers inside the cell and keep intracellular-free PLP concentration below toxic levels, but enough to saturate all PLP-dependent enzymes [57].

$B_6$ vitamers are absorbed from food and from the intestinal microflora. The richest sources of vitamin $B_6$ include fish, beef liver and other organ meats, potatoes and other starchy vegetables, and fruit. In animal-derived foods, vitamin $B_6$ is mainly present as PLP, associated with glycogen phosphorylase, and in smaller amounts as PMP, while in plants it is present as PN and PN-5′-β-D-glucoside [58]. Once ingested, PLP, PNP and PMP are dephosphorylated by the ecto-enzyme TNSALP. PM, PN and PL are absorbed from the upper small intestine by a carrier-mediated system and delivered through the portal circulation to the liver. In this organ, they are converted to PLP through the combined action of PDXK and PNPO. From the liver, PLP bound to albumin and dephosphorylated $B_6$ vitamers reach all tissues through the blood stream. In order to enter the cells, PLP needs to be dephosphorylated again by membrane-associated TNSALP. Membrane transporters of $B_6$ vitamers are yet to be identified. In the cytoplasm, PL, PN and PM are converted into the 5′-phosphorylated vitamers by PDXK, while PNPO converts PNP and PMP into PLP [59]. Once made available, PLP is somehow targeted to the dozens of different apo-$B_6$ enzymes that are being synthesized in the cell. Catabolism of vitamin $B_6$ consists in the oxidation of PL to 4-pyridoxic acid by aldehyde oxidase 1 (AOX-1) and NAD-dependent dehydrogenases [60].

## 2.2. Effects of vitamin $B_6$ homeostasis imbalance

The recommended dietary allowance of vitamin $B_6$ is less than 2 mg, an amount easily acquired in developed countries within any diet. PLP concentrations tend to be low in people with alcohol dependence [61], obese individuals [62] and pregnant women [63]. Some pathological conditions are associated with vitamin $B_6$ deficiency: end-stage renal diseases, chronic renal insufficiency and other kidney diseases [63]. In addition, vitamin $B_6$ deficiency can result from malabsorption syndromes, such as celiac disease, inflammatory bowel diseases including Crohn's disease and ulcerative colitis [63,64]. Certain genetic diseases, such as homocystinuria, can also cause vitamin $B_6$ deficiency [65]. People with rheumatoid arthritis often have low vitamin $B_6$ concentrations, and vitamin $B_6$ concentrations tend to decrease with increased disease severity [66]. Moreover, the assumption of certain drugs, such as contraceptives, and natural compounds may reduce PLP availability [67,68]. The symptoms of PLP deficiency determined by the above-mentioned conditions can be reverted by vitamin $B_6$ supplementation. It is known that vitamin $B_6$ supplements can also reduce the symptoms of premenstrual

syndrome [69], and are used to treat nausea and vomiting in pregnancy [70] as well as carpal tunnel syndrome [71]. Unfortunately, about 28–36% of the general population uses supplements containing vitamin $B_6$, even when unnecessary. It is important to maintain the correct balance of vitamin $B_6$ because several reports indicated that its excess is neurotoxic. Large doses of vitamin $B_6$ have detrimental effects (when the intake exceeds 200 mg day$^{-1}$), mostly evident at the level of the peripheral nervous system [59].

Importantly, perturbations of PLP homeostasis can also have genetic origins, determined by mutations in genes encoding proteins involved in vitamin $B_6$ metabolism, and causing severe neurological conditions (table 3). However, increasing evidence is accumulating that vitamin $B_6$ deficiency can also contribute to or be the main cause of the onset of serious diseases such as cancer and diabetes, as will be discussed in the following paragraphs.

# 3. Relationships between vitamin $B_6$, DNA damage and cancer inferred by epidemiological studies

## 3.1. Antioxidant properties of vitamin $B_6$

The antioxidant properties of $B_6$ vitamers were first recognized when it was discovered that the biosynthesis of vitamin $B_6$ is essential for the resistance of *Cercospora nicotianae* to singlet-oxygen-generating phototoxins [41]. The efficient activity of vitamin $B_6$ in quenching reactive oxygen species (ROS) was also demonstrated in plants [96]. Reduced levels of vitamin $B_6$ were associated with severe susceptibility to abiotic stress (oxidative, salt, drought, UVB) in plants, fungi and yeast [97]. Several studies demonstrated that the antioxidant properties of $B_6$ vitamers can derive from their direct involvement in reactions with ROS [42,98,99]. The strong antioxidant activity of $B_6$ vitamers originates from the presence of both hydroxyl (–OH) and amine (–NH$_2$) substituents on the pyridine ring, which can directly react with the peroxy radicals [100]. The antioxidant properties of vitamin $B_6$ also have an indirect cause and are surely linked to its role as enzyme cofactor. Studies on the radical-mediated oxidative damage in human whole blood demonstrated a surprising antioxidant activity of pyridoxine [101]. These observations may be attributed to the role of PLP as cofactor in the transulfuration pathway, in which homocysteine is converted to cysteine, a precursor of glutathione, a key regulator of intracellular redox state. It is known that vitamin $B_6$ and FD can lead to elevated homocysteine levels, which in turn generate ROS [102]. Also, the gasotransmitter $H_2S$ and taurine, involved in inflammation and chronic illnesses, derive from sulfur amino acids through the action of PLP-dependent enzymes [103]. The antioxidant properties of vitamin $B_6$ are also likely to be connected to its recognized role as anti-inflammatory agent, although a clear link between inflammation, $B_6$ status and carcinogenesis has not yet been established [103]. On the other hand, it has been demonstrated that $B_6$ vitamers are endogenous photosensitizers that enhance UVA-induced photooxidative stress in human skin. In particular, PL is the most phototoxic UVA-activated vitamer, probably because of the excited triplet state photochemistry associated with its aldehyde group [104].

**Table 3.** Inheritable diseases caused by PLP deficiency.

| name of disease and OMIM entry | gene involved in disease and name of encoded protein | affected metabolism | symptoms | available treatments | bibliography |
|---|---|---|---|---|---|
| PNPO deficiency (OMIM 610090) | *PNPO*; pyridoxine 5'-phosphate oxidase | PLP salvage pathway | severe neonatal/infantile seizures; few cases with onset after first year of life | pyridoxine/PLP supplementation | [72–77] |
| alkaline phosphatase deficiency (hypophosphatasia) according to age of onset: adult (OMIM 146300); perinatal (OMIM 241500); infantile (OMIM241500); childhood (OMIM 241510); odontohypophosphatasia (OMIM 146300) | *ALPL*; tissue-non-specific alkaline phosphatase | cellular uptake of $B_6$ vitamers | defective mineralization of bone and teeth; wide clinical spectrum, from stillbirth to fractures of the lower extremities or even no bone manifestations (odontohypophosphatasia) | pyridoxine/ pyridoxal supplementation | [78–86] |
| hereditary motor and sensory neuropathy, type VIC, with optic atrophy (OMIM 618511) | *PDXK*; pyridoxal kinase | PLP salvage pathway | progressive distal muscle weakness and atrophy of lower limbs; onset of neuropathy in the first decade, with difficulty of walking and running, followed by similar involvement of upper limbs and hands; distal sensory impairment; progressive optic atrophy and visual impairment during adulthood | PLP supplementation | [87] |
| PLP-binding protein deficiency (early-onset vitamin B6-dependent epilepsy) (OMIM 617290) | *PLPBP*; (pyridoxal phosphate-binding protein) | intracellular homeostatic regulation of PLP | onset of seizures in the neonatal period or first months of life | pyridoxine/PLP supplementation | [56] |
| pyridoxine-dependent epilepsy (α-aminoadipic semialdehyde dehydrogenase or antiquitin deficiency) (OMIM 266100) | *ALDH7A1* (α-aminoadipic semialdehyde dehydrogenase or antiquitin) | lysine degradation pathway; accumulation of pipecolic acid in plasma and cerebrospinal fluid | recurrent seizures in the prenatal, neonatal and postnatal period; few cases with onset after first year of life and adolescence | pyridoxine supplementation | [88–91] |
| L-$\Delta^1$-pyrroline-5-carboxylate dehydrogenase deficiency (Hyperprolinaemia type II) (OMIM 239510) | *ALDH4A1* (L-$\Delta^1$-pyrroline-5-carboxylate dehydrogenase) | proline degradation pathway; accumulation of proline and L-$\Delta^1$-pyrroline-5-carboxylic acid in plasma | often benign but clinical signs may include neonatal/infantile seizures; onset of seizures usually in infancy or childhood | pyridoxine supplementation | [92–95] |

## 3.2. Vitamin B$_6$ availability and cancer risk

The antioxidant properties of vitamin B$_6$ are expected to be beneficial in terms of cancer prevention and therapy. However, vitamin B$_6$ supplementation has been found to have controversial effects on tumour insurgence and progression [105]. In the attempt to interpret such controversial behaviour, it should be considered that PLP, being involved as cofactor in several biosynthetic pathways, is required for cell proliferation. Therefore, the availability of vitamin B$_6$ is bound to affect oncogenesis and tumour progression. Under this perspective, until the early 1980s, restricting vitamin B$_6$ availability was considered a promising therapeutic approach against cancer [105]. However, analyses performed on a large number of observations gave evidence of a strong inverse association between both vitamin B$_6$ dietary intake and PLP blood levels and cancer [106,107]. Vitamin B$_6$ deficiency is linked to a clear increase of several types of tumours, in particular affecting the gastrointestinal tract [108,109] and lungs [107]. Therefore, it is clear that vitamin B$_6$ has a complex and multifaceted role in cancer, as an antioxidant preventive agent, but also as an essential micronutrient required for cell proliferation. The low vitamin B$_6$ levels observed in cancer patients may be linked to the increased biosynthetic requirements of tumour cells and may also be partially responsible for their decreased immunity.

## 3.3. Expression of vitamin B$_6$ metabolism genes and cancer

Referring to the role of vitamin B$_6$ in cell proliferation, there is very strong evidence of an association between the expression of genes involved in the recycling of PLP and cancer.

The PNPO gene, encoding pyridoxine 5′-phosphate oxidase, is one out of seven genes, selected among 6487, whose altered expression was found to have a prognostic value in patients with colorectal cancer, and the expression of PNPO is increased in colorectal cancer tissues compared with adjacent normal tissues [110]. This is a clear evidence of the involvement of vitamin B$_6$ metabolism in cancer. Several other evidences have been reported, showing a link between salvage pathway enzymes and different kinds of tumours. Zhang *et al.* [111] demonstrated that PNPO contributes to the progression of ovarian surface epithelial tumours. Also in this case, PNPO was found to be overexpressed, and its knockdown induced cell apoptosis and decreased cell proliferation, migration and invasion *in vitro*. Moreover, silencing of PNPO inhibited tumour formation *in vivo* in orthotopically implanted nude mice. Interestingly, the same work also suggested that PLP is important for the regulation of PNPO expression, because PLP supplementation had the effect to suppress PNPO protein expression, resulting in the inhibition of epithelial ovarian cell proliferation. Furthermore, PNPO expression was shown to be regulated by the transforming growth factor-β, probably through the upregulation of a small RNA (miR-143-3p). PNPO is overexpressed also in breast invasive ductal carcinoma, where the expression level is inversely correlated with the overall survival of patients [112]. Moreover, knockdown of PNPO results in a decrease of breast cancer cell proliferation, migration, invasion and colony formation, arrests cell cycle at the G2/M phase and induces cell apoptosis. Also in this case, non-coding RNAs (MALAT1 and mir-216b-5p) are involved in PNPO regulation [112].

PDXK was demonstrated to be upregulated in non-small-cell lung cancer (NSCLC) [113]. Because the high protein levels of PDXK in the tumour did not correlate with the amounts of its mRNA, the authors suggested that PDXK expression is subjected to a post-translational control. Analogously, PDXK was recently found to be abundantly expressed in myeloid leukaemia cells, where PDXK depletion has an antiproliferative effect, which neither PN nor PM was able to rescue [114]. Consistently, pharmacological inhibition of PDXK using isoniazid or the more specific 4′-O-methylpyridoxine (gingkotoxin) has the same effect as genetic depletion of PDXK, suggesting that vitamin B$_6$ in plasma supports leukaemia proliferation [114]. On the other hand, it has been demonstrated that PDXK knockdown in human NSCLC cells protects against the cytotoxic activity of different agents, in particular the chemotherapy agent cisplatin, whereas PN administration improved, in a manner that depends on the presence of PDXK, cisplatin antitumour effect by exacerbating DNA damage. These observations were attributed to a pharmacokinetic effect of vitamin B$_6$, which favours the intracellular accumulation of cisplatin [115].

Taken together, these data suggest that the effect of vitamin B$_6$ on cancer should be examined from different points of view, according to the particular context that is taken under consideration, such as the protection from DNA damage, stimulation of immune response or cell proliferation (figure 3).

## 3.4. Impact of vitamin B$_6$ deficiency on DNA metabolism

Expectedly, vitamin B$_6$ deficiency plays an important role in DNA damage and repair. PLP is the cofactor of serine hydroxymethyltransferase (SHMT), whose folate-dependent reaction is the main source of one carbon units in metabolism, and plays a fundamental role in the synthesis of thymidylate (figure 1). PLP deficiency may determine a decrease in activity of SHMT, thereby causing the misincorporation of uracil in DNA [116–118]. Another enzyme that depends on both PLP and folate for its activity, glycine decarboxylase, a subunit of the glycine cleavage system, is fundamental for the synthesis of purines and therefore for DNA metabolism [119].

Given the implication of vitamin B$_6$ in DNA metabolism, it is not surprising that low vitamin B$_6$ levels have been associated with the formation of micronuclei in animal models [120] and in patients affected by inflammatory bowel disease [121]. Vitamin B$_6$ can impact on DNA also through different mechanisms. It has been proposed that vitamin B$_6$ suppresses endothelial cell proliferation and angiogenesis by inhibiting the activities of DNA polymerase and DNA topoisomerases [122]. In addition, in epatoma cells, PL induces the expression of the insulin-like growth factor-binding protein 1 via a mechanism involving the ERK/c-Jun pathway [123]. Moreover, the same vitamer was shown to play a role in increasing the expression of p21 via p53 activation in several cancer cells and mouse colon [124].

The proper uptake of vitamins is crucial to maintain genome stability, but it is important also to take into consideration the impact that an individual's genotype can have on the capacity to absorb, transport and metabolize vitamins. This could affect the intracellular level of vitamin B$_6$, which does not necessarily correspond to the level and distribution of plasmatic vitamin B$_6$.

An emerging body of research is focused on understanding how the genome affects folate metabolism and disease risk.

royalsocietypublishing.org/journal/rsob Open Biol. 10: 200034

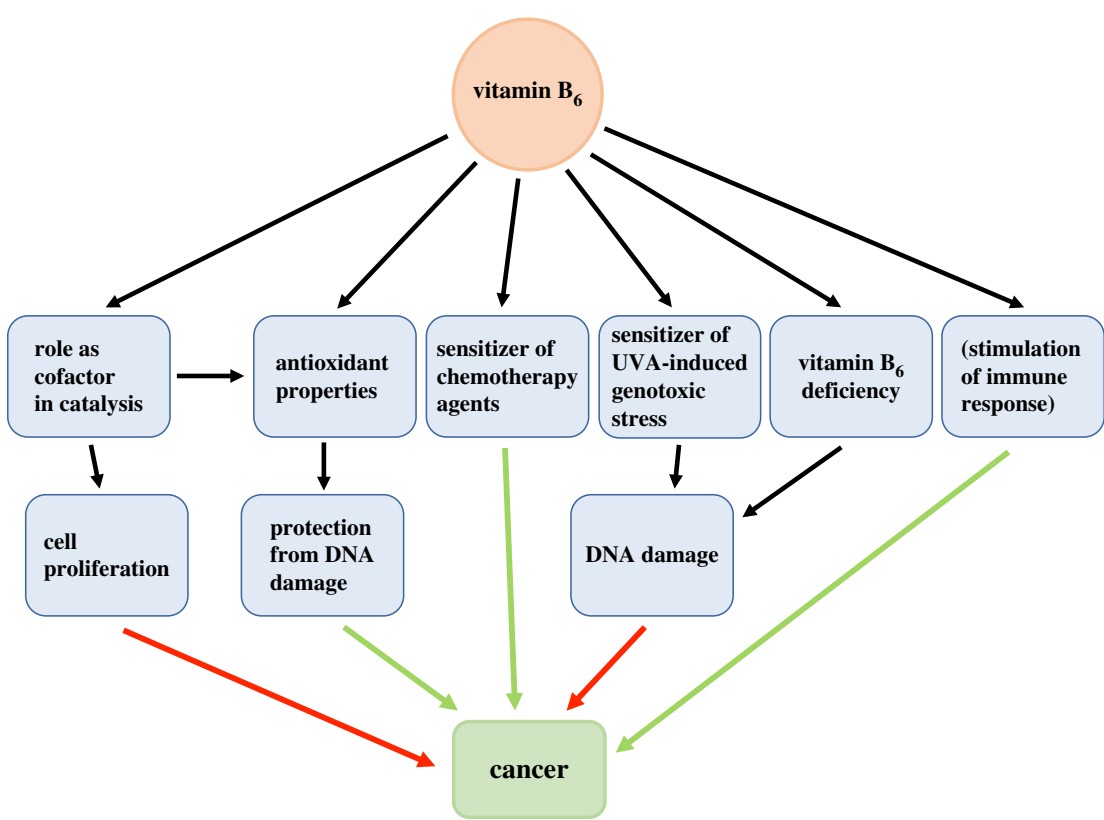

**Figure 3.** Relationships between vitamin B$_6$ and cancer. In the scheme, green arrows represent a protective effect against cancer, whereas red arrows indicate a promoting cancer effect.

The study of common polymorphisms in genes encoding for proteins required for folate metabolism (e.g. methylenetetrahydrofolate reductase (MTHFR; 677C > T), MS (MTR; 2756 A > G)) and uptake (e.g. glutamate carboxypeptidase II (GCPII; 1561 C > T), reduced folate carrier (RFC; 80 G > A)) revealed altered catalytic activity or expression of these proteins, suggesting that they can have a critical impact on developmental or progression of diseases [125,126]. Furthermore, since some of these enzymes for their function require other dietary cofactors (e.g. vitamin B$_2$ and B$_{12}$ are cofactors for MTHFR and MTR, respectively), it is important to consider not only nutrient-gene interactions but also interactions of folate with other nutrients. However, these kinds of studies performed *in vitro* or through human genetic screening suffer from the limitations imposed by these complex systems. To overcome these difficulties, the genetic approach applied to model organisms represents the best choice as it allows one to evaluate in whole organisms the phenotypic consequences elicited by mutations in genes involved in metabolism of specific vitamins. In this review, we show how this approach was successful in *Drosophila,* by providing novel approaches for determining the molecular mechanisms correlating micronutrients imbalance and cancer.

## 4. *Drosophila* as a model system to study the effects of B$_6$ depletion

Model organisms offer suitable contexts to study the physiopathology of many human diseases by overcoming the difficulties associated with human research. In this contest, *Drosophila melanogaster* has several advantages including an affordable maintenance, a short life cycle, a high fecundity, a relatively brief generation time, a well-characterized genome, a manageable number of chromosomes (consisting in a pair of

sex chromosomes along with three pairs of autosomes) and the availability of several mutant fly lines. Main metabolic molecular pathways are well conserved and about 75% of known human disease genes have related sequences in *Drosophila*. Thus, hypotheses and models generated using flies often prove to be relevant to biomedicine. In the last few years, *Drosophila* has been considered a precious model for several metabolic diseases including diabetes and has acquired interest for nutritional intervention studies. The impacts of diet on lifespan, locomotor activity, intestinal barrier function and gut microbiota, as well as fertility, have been evaluated in order to investigate diet-induced pathophysiological mechanisms including inflammation and stress response [127]. However, so far, only a few papers in the literature report studies on the role of vitamins in *Drosophila*. Bahadorani *et al.* [128] studied antioxidant and pro-oxidant properties of vitamin A, C and E to *Drosophila* lifespan under normoxia and oxidative stress. Other investigations [129,130] focused on the role of fly microbiota in providing essential folates and vitamin B$_1$ to their host when those are scarce in the diet. Another study [131] showed that folate supplementation was able to alleviate mitochondrial dysfunction in a Parkinson fly model. However, to our knowledge, only vitamin B$_6$ has been studied in detail in a *Drosophila* model with the aim to understand cellular and molecular mechanisms at the basis of its beneficial effect on human diseases [132–135].

### 4.1. Impact of mutations in *PDXK* and *PNPO* genes, involved in vitamin B$_6$ activation, on genome stability

Like mammals, *Drosophila* produces PLP through the salvage pathway, by recycling precursors from food. In the standard food where flies grow, vitamin B$_6$ is present in brewer's yeast,

which is rich in PM and PN but poor in PL [136]. As a consequence, it is expected that PDXK and PNPO depletion shall cause similar phenotypes by blocking PLP synthesis.

Molecular mechanisms at the basis of vitamin $B_6$ metabolic functions have been investigated in detail by examining phenotypes elicited by mutations ($dPdxk^1$ and $dPdxk^2$) in PDXK encoding gene. Cytological analysis revealed 5% of chromosome aberrations (CABs) in larval brain cells from $dPdxk$ mutants (in wild-type brains, CABs frequency ranges from 0.3 to 0.5% [137]). CABs, such as chromatid deletions, isochromatid deletions and chromosome exchanges, were completely rescued by PLP supplementation (1 mM), suggesting that PLP is important for chromosome integrity maintenance. As a confirmation of this, wild-type larvae treated with PLP inhibitors, such as 4-deoxypiridoxine (4-DP), cycloserine, penicyllamine and isoniazid, also displayed high CAB frequency [132]. Interestingly, most of these compounds are currently used as drugs for several human diseases, such as depression, arthritis and tuberculosis, which have the side effect of decreasing PLP levels [65].

In *Drosophila*, the counterpart of PNPO enzyme is encoded by *Sgll* gene [138]. The depletion of this enzyme causes epilepsy in human, *Drosophila* and zebrafish [139–141]. Silencing of *Sgll* by RNA interference produced 3% of CABs, and was rescued not only by PLP but also by PL supplementation [135].

Data obtained in *Drosophila* are in line with those obtained in yeast, in which it has been demonstrated that mutations in *BUD16* gene (the gene encoding PDXK in yeast) induce gross chromosome rearrangements [142]. Furthermore, also in human cells, PDXK depletion and 4-DP treatment in mock cells cause CABs [132] and 53BP1 repair foci [142], confirming the role of PLP in genome integrity maintenance also in higher organisms, consistently with the presence of micronuclei found in cells with low $B_6$ levels [120,121].

## 4.2. Mechanisms through which vitamin $B_6$ protects from DNA damage

Vitamin $B_6$, as well as folate and vitamin $B_{12}$, are involved in the one carbon metabolism, a crucial pathway for DNA synthesis and repair (figure 1). In particular, vitamin $B_6$ serves as coenzyme for the activity of SHMT, which directs one carbon units towards thymidylate synthesis. HPLC analysis revealed that *dPdxk* mutants display increased dUTP/dTTP ratio, but they do not show increased sensitivity to hydroxyurea (HU), a drug which interferes with replication process [132]. This suggests that replication failure is not at the basis of the CABs in *dPdxk* mutants. However, as nucleotide imbalance also affects DNA repair, it is possible that it may contribute to CABs. By contrast, in yeast, HU dramatically affects the growth of *bud16Δ* mutant cells, thus it has been hypothesized that PLP deficiency triggers DNA lesions due to a nucleotide imbalance [142].

Studies in *Drosophila* revealed that hyperglycaemia induced by low PLP levels might represent another potential cause of CABs. In fact, besides CABs, *dPdxk* mutants display increased glucose content in larval haemolymph in part due to insulin resistance [132], a metabolic condition at the basis of type 2 diabetes. Diabetic hallmarks are also evident in flies fed with 4-DP and in flies depleted of Sgll, which also exhibit impaired lipid metabolism and small body size, a typical feature of diabetic flies [133,135]. The hypothesis that high

**Figure 4.** Effects of vitamin $B_6$ deficiency inferred from studies carried out in *Drosophila*.

glucose can produce CABs in low PLP contexts came from two observations. *dPdxk* mutants, Sgll-depleted larvae and 4-DP-fed larvae grown in a medium supplemented with sugars (sucrose or glucose or fructose) exhibit a further increase in CABs (ranging from 15 to 60%), differently from wild-type larvae in which sugar treatment leaves unchanged CAB frequency [132,135]. In addition, *dPdxk* mutants, Sgll-depleted flies and 4-DP-fed larvae accumulated high concentrations of advanced glycation end products (AGEs) in brains [132,133,135]. In high-glucose conditions, these molecules originate from non-enzymatic glycation of amino groups of proteins and DNA and are genotoxic due to ROS formation [143]. AGEs have been associated with diabetic complications and are quenched by PLP and PM [51,52]. Interestingly, α-lipoic acid, a compound able to decrease AGE formation, rescues not only AGEs but also CABs in brains from *dPdxk*$^1$ mutants, Sgll-depleted individuals and 4-DP-fed larvae [132,135]. Taken together, these findings suggested that in low PLP conditions, CABs are mostly produced by hyperglycaemia, which in turn promotes AGE accumulation that causes DNA damage [132,135] (figure 4). Studies on human cells confirmed this model, as in HeLa cells depleted for PDXK enzyme glucose treatment increases CABs and lipoic acid is effective in rescuing them [132]. Interestingly, a combined effect of low vitamin levels and high glucose in inducing DNA damage has also been found for folates in human cell lines [144].

As mentioned above, some studies indicated the existence of a correlation between low PLP levels and cancer. Although mechanisms behind this association are not clear, it has been hypothesized that low PLP levels can impact on cancer through different mechanisms, for example, by increasing inflammation, decreasing immune defences and promoting genome instability [105]. The finding obtained in *Drosophila* not only confirmed the hypothesis that low PLP levels increase cancer risk through DNA damage, but also revealed that DNA damage in PLP-deficient cells can in part be due to AGE accumulation, which adds to our knowledge of the complex relationship between vitamin $B_6$ and cancer.

## 4.3. Vitamin B$_6$ as a potential link between diabetes and cancer

Recent data obtained in *Drosophila* suggested that low PLP levels may increase cancer risk in diabetic patients, providing a mechanistic link between studies in humans that associate PLP with cancer and studies indicating that diabetic patients have a higher risk of developing various types of cancer [145–147]. It has been shown that treatment with vitamin B$_6$ antagonist 4-DP resulted in much more severe DNA damage in diabetic individuals than in wild-type flies. Brains from two different models of type 2 diabetes displayed 60–80% of CABs (versus 25% in wild-type) and accumulated many more AGEs. Moreover, double mutants bearing $dPdxk^1$ mutation which abolishes PLP production and $Akt1^{04226}$ mutation which impairs insulin signalling showed a synergistic interaction in CABs formation [133]. It is well known that diabetic condition increases oxidative stress and impairs DNA repair [148]. Accordingly, oxidative damage and DNA strand breaks have been found in both type 1 and type 2 diabetic patients [149,150]. Thus, in a diabetic context, PLP deficiency enhances genome instability by producing a further weakening of antioxidant defence and enhancing hyperglycaemia, contributing to DNA damage throughout ROS induced by AGEs. Since CABs are strictly linked to cancer development and/or progression, extrapolated to humans, these data indicate that low PLP levels may represent a cancer risk factor for diabetic patients. This finding is particularly relevant because the diabetic condition *per se* lowers PLP levels in animal models and patients [151]. Moreover, these data reinforce the hypothesis that besides inflammation, hyperinsulinaemia and hyperglycaemia, DNA damage plays an important role in driving diabetic cells towards malignant transformation.

## 4.4. Validation in *Drosophila* of PDXK human variants and their impact on chromosome integrity

*Drosophila* is also a useful means of validating the causative nature of candidate genetic variants found in patients, and of obtaining functional information on the relationship between disease and linked gene [152]. This approach has been employed to further confirm the role of *PDXK* human gene in chromosome integrity maintenance and to strengthen the model in which CABs are largely produced by hyperglycaemia in low PLP conditions [134]. From these studies it emerged that the expression

in $dPdxk^1$ flies of four PDXK variants (three—D87H, V128I and H246Q—listed in databases, and one—A243G—found in a genetic screening in patients with diabetes) was unable to rescue CABs, hyperglycaemia and AGE accumulation, differently from PDXK wild-type protein. Moreover, biochemical analysis of D87H, V128I, H246Q and A243G mutant proteins revealed reduced catalytic activity and reduced affinity for B$_6$ vitamers, giving an explanation for this behaviour. Although these variants are rare in population and carried in heterozygous condition, these findings suggest that in certain metabolic contexts and diseases in which PLP levels are reduced, the presence of these PDXK variants could threaten genome integrity and contribute to increased cancer risk.

## 5. Conclusion

B group vitamins are crucial compounds for human health, as they have a strong impact on genome stability and cancer. The relationship between vitamin B$_6$ and cancer, deduced from studies reported in this review, is complex and leads us to speculate that it can result from a balance between its antioxidant properties on the one hand and its role as a micronutrient important for cell metabolism on the other hand.

As described in this review, *D. melanogaster* turned out to be a precious model for this kind of study. Findings obtained in *Drosophila* provided information regarding the mechanisms at the basis of the impact of vitamin B$_6$ on DNA damage, revealing that AGEs can play an important role. In addition, they suggest that low vitamin B$_6$ levels could represent a cancer risk factor in diabetes patients. Future studies in this model organism will be useful to further deepen knowledge of the mechanisms by which vitamin B$_6$ and other vitamins can protect against DNA damage and cancer, with the aim of developing personalized treatments.

Data accessibility. This article does not contain any additional data.

Authors' contributions. All authors have contributed to the writing of the manuscript.

Competing interests. We declare we have no competing interests.

Funding. Research was supported by grants from Sapienza (Progetto di Ateneo, to F.V. and R.C.), from Istituto Pasteur Italia-Fondazione Cenci Bolognetti (Research grant 'Anna Tramontano' 2018, to R.C.), from Consiglio Nazionale delle Ricerche Italy-Russia bilateral project (CUP B86C17000270001, to A.T.) and from Russian Foundation for Basic Researches (grant no. 18-54-7812, to V.B.).

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
