## [Reviewer comments · Open Biology]

Review History

RSOB-20-0034.R0 (Original submission)

Review form: Reviewer 1

Recommendation

Accept as is

Do you have any ethical concerns with this paper?

No

Comments to the Author

The review article from Contestabile and collaborators illustrates the antioxidant properties of vitamin B₆ and the role of Vitamin B₆ metabolism in human diseases with a special attention to cancer. Moreover the article reports the advantages offered by *Drosophila* as a model system in studying the molecular mechanisms underlying the involvement of Vitamin B₆ in DNA damage and cancer. I think the review is very well-structured and well-written.

I think it should be accepted as it is.

Review form: Reviewer 2

Recommendation

Accept with minor revision (please list in comments)

Do you have any ethical concerns with this paper?

No

Comments to the Author

The paper by Contestabile R. et al. describes the metabolism of Vitamin B6 and its role on a variety of human diseases, with particular focus on the effects related to DNA damage and on the use of *Drosophila* as model organism. Overall, the paper is clearly written, although few examples of repetitive sentences are present, and the relevance of the topic for a general audience is high. Therefore, it is recommended to publish provided a few points are clarified.

-Page 4. The dietary sources of Vitamins B9, B12 and B1 should be cited. Since folates are vitamins, the sentence beginning with "Humans cannot synthesize folate ..." can be removed.

-Page 8, lines 3-10. Please add the citation of bibliographic sources.

-Page 10, line 7. I would suggest to substitute "synthesis" with "recycling"

-Page 11, chapter 3.4. Please change "DNA of polymerase" with "of DNA polymerase"

-Page 12 The authors should consider the possibility that the levels and distribution of plasmatic and intracellular Vitamin B6 forms do not necessarily correspond.

-Page 15, chapter 4.3. The relationship between PLP levels and cancer/diabetes should be better clarified.

Decision letter (RSOB-20-0034.R0)

02-Mar-2020

Dear Dr Verni,

We are pleased to inform you that your manuscript RSOB-20-0034 entitled "The multifaceted role of vitamin B6 in cancer: *Drosophila* as a model system to investigate DNA damage" has been accepted by the Editor for publication in *Open Biology*. The reviewer(s) have recommended publication, but also suggest some minor revisions to your manuscript. Therefore, we invite you to respond to the reviewer(s)' comments and revise your manuscript.

Please submit the revised version of your manuscript within 7 days. If you do not think you will be able to meet this date please let us know immediately and we can extend this deadline for you.

When submitting your revised manuscript, you will be able to respond to the comments made by the referee(s) and upload a file "Response to Referees" in "Section 6 - File Upload". You can use this to document any changes you make to the original manuscript. In order to expedite the

processing of the revised manuscript, please be as specific as possible in your response to the referee(s).

- 1) A text file of the manuscript (doc, txt, rtf or tex), including the references, tables (including captions) and figure captions. Please remove any tracked changes from the text before submission. PDF files are not an accepted format for the "Main Document".
- 2) A separate electronic file of each figure (tiff, EPS or print-quality PDF preferred). The format should be produced directly from original creation package, or original software format. Please note that PowerPoint files are not accepted.
- 3) Electronic supplementary material: this should be contained in a separate file from the main text and meet our ESM criteria (see <http://royalsocietypublishing.org/instructions-authors#question5>). All supplementary materials accompanying an accepted article will be treated as in their final form. They will be published alongside the paper on the journal website and posted on the online figshare repository. Files on figshare will be made available approximately one week before the accompanying article so that the supplementary material can be attributed a unique DOI.

Online supplementary material will also carry the title and description provided during submission, so please ensure these are accurate and informative. Note that the Royal Society will not edit or typeset supplementary material and it will be hosted as provided. Please ensure that the supplementary material includes the paper details (authors, title, journal name, article DOI). Your article DOI will be 10.1098/rsob.2016[*last 4 digits of e.g. 10.1098/rsob.20160049*].

- 4) A media summary: a short non-technical summary (up to 100 words) of the key findings/importance of your manuscript. Please try to write in simple English, avoid jargon, explain the importance of the topic, outline the main implications and describe why this topic is newsworthy.

Images

Data-Sharing

It is a condition of publication that data supporting your paper are made available. Data should be made available either in the electronic supplementary material or through an appropriate repository. Details of how to access data should be included in your paper. Please see <http://royalsocietypublishing.org/site/authors/policy.xhtml#question6> for more details.

Data accessibility section

Sincerely,
The Open Biology Team
mailto:openbiology@royalsociety.org

Reviewer(s)' Comments to Author:

Referee: 1

Comments to the Author(s)

The review article from Contestabile and collaborators illustrates the antioxidant properties of vitamin B6 and the role of Vitamin B6 metabolism in human diseases with a special attention to cancer. Moreover the article reports the advantages offered by *Drosophila* as a model system in studying the molecular mechanisms underlying the involvement of Vitamin B6 in DNA damage and cancer. I think the review is very well-structured and well-written. I think it should be accepted as it is.

Referee: 2

Comments to the Author(s)

The paper by Contestabile R. et al. describes the metabolism of Vitamin B6 and its role on a variety of human diseases, with particular focus on the effects related to DNA damage and on the use of *Drosophila* as model organism. Overall, the paper is clearly written, although few examples of repetitive sentences are present, and the relevance of the topic for a general audience is high. Therefore, it is recommended to publish provided a few points are clarified.

-Page 4. The dietary sources of Vitamins B9, B12 and B1 should be cited. Since folates are vitamins, the sentence beginning with "Humans cannot synthesize folate ..." can be removed.

-Page 8, lines 3-10. Please add the citation of bibliographic sources.

-Page 10, line 7. I would suggest to substitute "synthesis" with "recycling"

-Page 11, chapter 3.4. Please change "DNA of polymerase" with "of DNA polymerase"

-Page 12 The authors should consider the possibility that the levels and distribution of plasmatic and intracellular Vitamin B6 forms do not necessarily correspond.

-Page 15, chapter 4.3. The relationship between PLP levels and cancer/diabetes should be better clarified.

Author's Response to Decision Letter for (RSOB-20-0034.R0)

See Appendix A.

Decision letter (RSOB-20-0034.R1)

03-Mar-2020

Dear Dr Verni,

We are pleased to inform you that your manuscript entitled "The multifaceted role of vitamin B6

in cancer: *Drosophila* as a model system to investigate DNA damage" has been accepted by the Editor for publication in Open Biology.

Article processing charge

Please note that the article processing charge is immediately payable. A separate email will be sent out shortly to confirm the charge due. The preferred payment method is by credit card; however, other payment options are available.

Sincerely,

The Open Biology Team

mailto: openbiology@royalsociety.org

Appendix A

RESPONSE TO REFEREE 2

QUESTION. Page 4. The dietary sources of Vitamins B9, B12 and B1 should be cited.

ANSWER. We added a new table (Table 1), which indicates the dietary sources.

QUESTION. Since folates are vitamins, the sentence beginning with “Humans cannot synthesize folate ...” can be removed.

ANSWER. The sentence was removed.

QUESTION. Page 8, lines 3-10. Please add the citation of bibliographic sources.

ANSWER. All the citations were added.

QUESTION. Page 10, line 7. I would suggest to substitute “synthesis” with “recycling”

ANSWER. The correction was done.

QUESTION. Page 11, chapter 3.4. Please change “DNA of polymerase” with “of DNA polymerase”

ANSWER. The correction was done.

QUESTION. Page 12 The authors should consider the possibility that the levels and distribution of plasmatic and intracellular Vitamin B6 forms do not necessarily correspond.

ANSWER. Thank you for your suggestion. We added a sentence to clarify this point.

QUESTION. Page 15, chapter 4.3. The relationship between PLP levels and cancer/diabetes should be better clarified.

ANSWER. We added 3 more sentences and bibliographic references in order to better clarify this point.